# Nuclear p62/SQSTM1 facilitates ubiquitin-independent proteasomal degradation of BMAL1

Chenliang Zhang[1☉*], Quanyou Wu[2☉], Huan Zhang[3☉], Ruichen Liu[4], Liping Li[5]

**1** Division of Abdominal Tumor Multimodality Treatment, Department of Medical Oncology, Cancer Center and Laboratory of Molecular Targeted Therapy in Oncology, West China Hospital, Sichuan University, Chengdu, Sichuan Province, China, **2** Division of Abdominal Tumor Multimodality Treatment, Department of Medical Oncology, Cancer Center, West China Hospital, Sichuan University, Chengdu, Sichuan Province, China, **3** Division of Thoracic Tumor Multimodality Treatment, Cancer Center, West China Hospital, Sichuan University, Chengdu, Sichuan Province, China, **4** West China School of Medicine, Sichuan University, Chengdu, Sichuan Province, China, **5** Department of Pharmacy, Chengdu Fifth People's Hospital, Chengdu University of Traditional Chinese Medicine, Chengdu, Sichuan Province, China

☉ These authors contributed equally to this work.
* zhangchenliang@wchscu.edu.cn

## Abstract

Brain and muscle arnt-like protein 1(BMAL1) is a critical regulator of circadian rhythm. Although transcriptional regulation of BMAL1 has been extensively studied, the mechanisms governing the stability of BMAL1 at the protein level remain unclear. p62/SQSTM1 is a crucial factor in proteostasis regulation and is involved in both autophagy and the ubiquitin-proteasome system. We demonstrated that p62 promotes proteasomal degradation of BMAL1 within the nucleus, independent of ubiquitination. Additional molecular analyses indicated that p62 functions as a receptor for the 20S proteasome, facilitating the recruitment of BMAL1 to the 20S proteasome for degradation. This mechanism is independent of recently identified p62-driven nuclear biomolecular condensates. We also revealed that remodeling the nuclear accumulation of p62 may represent a potential strategy for targeting BMAL1 to suppress tumor cell growth. In conclusion, our findings revealed a novel mechanism by which nuclear p62 regulates BMAL1 proteostasis.

## Author summary

p62/SQSTM1 is a well-established regulator of protein quality control, playing a key role in the autophagic degradation of various proteins. Furthermore, studies have shown that p62 can promote the proteasomal degradation of ubiquitinated proteins. Recent research has highlighted that p62 drives the proteasomal degradation of nuclear proteins in a manner dependent on both ubiquitin and biomolecular condensates. In this study, we discovered that nuclear p62 promotes the proteasomal degradation of BMAL1, a critical regulator of the circadian rhythm,

**Data availability statement:** All relevant data are contained within the main manuscript or supporting information files.

**Funding:** This work was supported by the Sichuan Science and Technology Program (grant number: 2024YFFK0343) to CLZ and the National Natural Science Foundation of China (grant numbers: 32000533) to CLZ. The funders had no role in study design, data collection and analysis, decision to publish, or preparation of the manuscript.

**Competing interests:** The authors have declared that no competing interests exist.

in a ubiquitin- and condensates-independent manner. Further analysis revealed that p62 acts as a bridge, linking BMAL1 to the 20S proteasome, thereby facilitating its proteasomal degradation. Additionally, we propose that modulating p62's ability to accumulate in the nucleus could serve as a strategy to inhibit BMAL1-associated tumor proliferation. Our findings uncover a novel mechanism through which BMAL1 proteostasis is regulated by nuclear p62, shedding light on a new aspect of p62's role in protein quality control.

## Introduction

The proteasome, a multimeric complex comprising numerous subunits, regulates cellular proteostasis by degrading proteins into short peptides or amino acids. It consists of two primary components: A 20S core particle, which exhibits proteolytic activity, and a 19S regulatory particle [1]. Together, the 20S and 19S proteasomes form the 26S proteasome, responsible for the degradation of ubiquitinated proteins. In this process, the 19S subunit recognizes and binds to the ubiquitin chain on substrates, directing them into the proteolytic channel of the 20S proteasome for degradation, a process known as the ubiquitin-proteasome system (UPS) [1,2]. In addition to the ubiquitin-dependent pathway, the proteasome can degrade certain oxidized proteins or intrinsically disordered proteins (IDPs), which possess one or more intrinsically disordered regions (IDRs) characterized by low sequence complexity and an inability to adopt stable tertiary structures, through a ubiquitin-independent mechanism, generally executed by the 20S proteasome alone, without the involvement of the 19S particle [3].

p62, also known as sequestosome 1 (SQSTM1), is a renowned regulator of protein quality control, mediating the autophagic degradation of various proteins. Structurally, p62 contains an N-terminal Phox and Bem1 (PB1) domain that facilitates homotypic or heterotypic oligomerization of PB1 domain-containing proteins, a C-terminal ubiquitin association (UBA) domain that recognizes and binds to ubiquitin chains on ubiquitinated proteins, and a central microtubule-associated protein light chain 3 (LC3)-interacting region (LIR) motif that binds to LC3 [4]. This protein recruits autophagosomal membranes after conjugation to phosphatidylethanolamine [5]. As an autophagy receptor, p62 presents ubiquitinated cargos to the autophagosome and promotes their degradation through the autophagosome-lysosome pathway [4]. Additionally, p62 contains nuclear localization signal (NLS) and nuclear export signal (NES) motifs, enabling nuclear import and export, respectively [6]. Recent studies have demonstrated that p62 can drive the proteasomal degradation of nuclear proteins in a ubiquitin- and biomolecular condensate-dependent manner [7,8].

Brain and muscle arnt-like protein 1(BMAL1) is a crucial controller of the circadian rhythm. As a transcription factor, BMAL1 forms a complex with another circadian protein, Circadian locomoter output cycles protein kaput (CLOCK), which activates the transcription of other circadian genes by binding to the E-box (CTGCAG) sequence, including cryptochrome (Cry) genes (*CRY1/2*) and Period (Per) genes (*PER1/2/3*)

[9,10]. The CRY and PER proteins, which accumulate in the cell, interact with the BMAL1/CLOCK complex and inhibit its transcriptional activity [9,10]. This interaction forms a 24-h regulatory loop in which BMAL1/CLOCK and their target circadian genes collectively govern the organism's circadian rhythm [9,10]. Dysregulation of BMAL1 expression and function has been implicated in various diseases, including neurodegenerative disorders [11], chronic airway diseases [12], diabetes [13], and cancer [10]. Although multiple studies have reported that BMAL1 can be degraded through UPS and autophagy pathways, the underlying molecular mechanisms, particularly regarding the stability of the BMAL1 protein within the nucleus, remain incompletely understood.

This study revealed that nuclear p62 promotes the recruitment of BMAL1 to the 20S proteasome and induces proteasomal degradation in a ubiquitin-independent manner. Furthermore, we suggest that remodeling p62's ability to accumulate in the nucleus can inhibit BMAL1-associated tumor proliferation. Our findings identified a novel mechanism by which BMAL1 proteostasis is regulated by nuclear p62.

## Results

### p62 promotes the proteasomal degradation of BMAL1

To investigate the influence of p62 on the protein stability of BMAL1, we co-expressed FLAG-tagged p62 and Myc-tagged BMAL1 in HEK293 and HeLa cells. The co-expression of p62 significantly reduced the exogenous BMAL1 levels (Fig 1A–1D). Subsequent analysis of the effects of the proteasome inhibitor MG132 and autophagy inhibitors (Bafilomycin A1 and Pep/EQ64) on p62-mediated BMAL1 degradation demonstrated that MG132 effectively inhibited BMAL1 degradation, whereas the autophagy inhibitors had no effect (Fig 1A–1D). This suggests that p62 may promote proteasomal degradation of BMAL1. To ascertain if p62 destabilizes the BMAL1 protein, we employed cycloheximide to block protein synthesis and monitored protein levels at various treatment times. Co-expression of p62 significantly increased the rate of BMAL1 degradation, which was inhibited by proteasome inhibitors (Fig 1E and 1F). Subsequently, we examined the stability of endogenous BMAL1 and the regulatory effects of p62 on it. In HEK293 cells, proteasome inhibition (but not autophagy) significantly elevated nuclear BMAL1 protein levels without affecting cytosolic BMAL1 levels (Fig 1G–1I). Furthermore, we observed that the knockout of p62 significantly increased the nuclear BMAL1 levels in AD293 cells (Fig 1J–1L). These findings indicated that p62 destabilizes nuclear BMAL1 by promoting proteasomal degradation. Given that BMAL1 is a critical regulator of circadian rhythm, we investigated the expression of clock genes in p62 deletion cells after synchronized by dexamethasone. Consistent with the destabilizing effect of p62 on BMAL1, the deletion of p62 increased the levels of several BMAL1-targeted clock transcripts, including *NR1D1*, *PER2*, *CRY1*, and *CRY2*, but not *PER1* and *DBP*, and decreased the transcription levels of *BMAL1* and *CLOCK* (Fig 1M–1T). To determine whether p62 regulates the expression of clock genes through BMAL1, we silenced *BMAL1* in p62 KO cells and examined its effect on clock gene expression. The results showed that silencing *BMAL1* reversed the increase of clock genes caused by p62 knockout (S1A and S1B Fig). This confirmed that p62 regulates the expression of clock genes by regulating BMAL1 protein levels.

### p62 drives BMAL1 degradation in the nuclear

Since p62 can promote protein degradation in the cytoplasm and nucleus, we investigated the specific subcellular localization of BMAL1 degradation mediated by p62. First, we examined the effects of different p62 mutants, exhibiting distinct cellular distributions, on BMAL1 protein stability. We observed that the p62 mutant lacking NLS (ΔNLS) abolished the regulatory effect of p62 on exogenous BMAL1 stability (Fig 2A, 2B, and 2E), whereas the p62 mutant (ΔNES) lacking NES enhanced exogenous BMAL1 degradation (Fig 2C–2E). Consistently, we also observed that re-expression of p62 (WT) or p62 (ΔNES) in p62 KO cells, rather than p62 (ΔNLS), significantly reduced the endogenous BMAL1 protein levels (S2A and S2B Fig). Furthermore, we assessed BMAL1 degradation mediated by p62 in the presence of leptomycin B (LMB), a well-known inhibitor of Chromosomal Maintenance 1 (CRM1)-mediated nuclear export. Our results indicated that, regardless of LMB treatment, co-expression of p62 significantly reduced BMAL1 protein levels, suggesting that p62 regulates

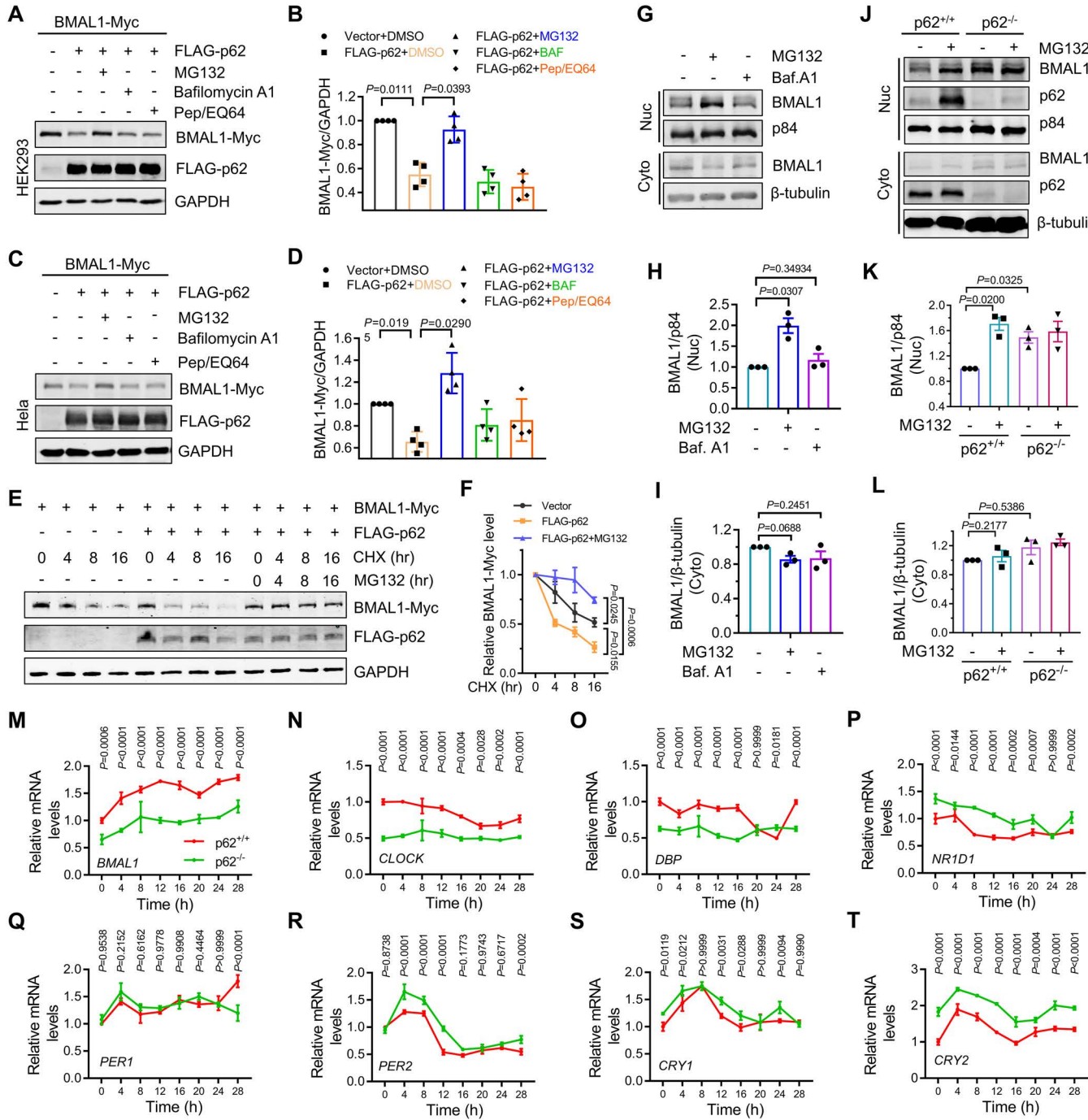

**Fig 1. p62 promotes the proteasomal degradation of BMAL1.** (**A**) HEK293 cells were transfected with the indicated plasmids. After 24 h, the cells were treated with the specified drugs (MG132 (2 µM), Bafilomycin A1 (20 nM), Pepstatin A (Pep, 25 µg/ml)/E64d (25 µg/ml)) for 16 h. The whole-cell lysates were subjected to western blot analysis with indicated antibodies. (**B**) Quantitative analysis of results in (**A**). Data are mean ± SEM of biological replicates (*n* = 4). (**C**) Hela cells were transfected with the indicated plasmids. After 24 h, the cells were treated with the specified drugs (MG132 (2 µM), Bafilomycin A1 (20 nM), Pepstatin A (Pep, 25 µg/ml)/E64d (25 µg/ml)) for 16 h. The whole-cell lysates were subjected to western blot analysis with indicated antibodies. (**D**) Quantitative analysis of results in (**C**). Data are mean ± SEM of biological replicates (*n* = 4). (**E**) HEK293 cells were transfected with the indicated plasmids. After 24 h, the cells were treated with the specified drugs (Cycloheximide (CHX, 25 µg/ml), MG132 (2 µM)) for varying durations. The whole-cell lysates were subjected to western blot analysis with indicated antibodies. (**F**) Quantitative analysis of results in (**E**). Data are mean ± SEM of biological replicates (*n* = 3). (**G**) HEK293 cells treated with MG132 (2 µM) or Bafilomycin A1 (20 nM) for 16 h. The cytosolic and nuclear fractions were

subjected to western blot analysis with indicated antibodies. (**H-I**) Quantitative analysis of results in (**G**). Data are mean ± SEM of biological replicates (*n* = 3). (**J**) Wild-type AD293 cells (p62$^{+/+}$) and p62 knockout AD293 cells (p62$^{-/-}$) treated with MG132 (2 μM) for 16 h. The cytosolic and nuclear fractions were subjected to western blot analysis with indicated antibodies. (**K-L**) Quantitative analysis of results in (**J**). Data are mean ± SEM of biological replicates (*n* = 3). (**M-T**) p62$^{+/+}$ and p62$^{-/-}$ cells were treated with Dexamethasone (100 μM) for 2 h. The mRNA levels of clock genes were then analysis at indicated time. Data are mean ± SEM of biological replicates (*n* = 3). For **B** and **D**, the *P* value was determined by a one-way ANOVA analysis. For **F**, the *P* value was determined by a one-way ANOVA analysis at 16 h. For **H**, **I**, **K**, and **L**, the *P* value was determined by Student's t test (two-sided). For **M-T**, the *P* value was determined by a two-way ANOVA analysis.

the stability of BMAL1 in the nucleus (Fig 2F–2J). Additionally, we found that LMB can reduce the levels of endogenous BMAL1 protein in cells (Figs 2K and 2L, and S2C and S2D), and this reduction could be reversed by proteasome inhibition (Fig 2K and 2L). These findings indicated that p62 facilitates proteasomal degradation of BMAL1 in the nucleus.

## p62 destabilizes BMAL1 through ubiquitin-independent proteasomal degradation

Previous studies have reported that p62 recruits ubiquitinated substrates to nuclear condensates and facilitates their degradation by the proteasome in a ubiquitin-dependent manner [7,8]. Our findings indicated that p62 does not promote BMAL1 ubiquitination (Fig 3A). Immunostaining results failed to reveal the co-localization of BMAL1 and p62-associated condensates, which exhibited significant enrichment of ubiquitin (S3A–S3C Fig). Although we observed that the deletion of the UBA domain that is primarily responsible for binding to ubiquitin chains abolished p62-mediated BMAL1 degradation (S3D and S3E Fig), neither the ubiquitination inhibitor PYR-41 nor the deubiquitination inhibitor PR-619 affected the reduction of BMAL1 induced by co-expression with p62 (Fig 3B and 3C), suggesting that the ubiquitination state of BMAL1 may not be essential for its degradation. To further validate the proteasomal degradation of BMAL1, independent of its ubiquitination state, we knocked down PSMD11 (S2A Fig), a subunit of the 19S proteasome that is crucial for ubiquitin-dependent proteasomal degradation [14,15]. Our findings revealed that silencing *PSMD11* did not mitigate p62-mediated destabilization of BMAL1 (Figs 3D and 3E and S3F), indicating that p62 promotes BMAL1 degradation through a ubiquitin-independent mechanism. IDPs can undergo degradation by the proteasome in a ubiquitin-independent manner [3]. Therefore, we examined the composition of BMAL1 IDPs to determine their contribution to BMAL1 degradation. Predictor of Natural Disordered Regions (PONDR) analysis and three-dimensional structure predictions revealed the presence of four disordered regions (DRs) within BMAL1: DR1 (AA 1–71), DR2 (AA 209–277), DR3 (AA 445–491), and DR4 (AA 512–590) (Fig 3F and 3G). Deleting any of these DRs abolished the phenotype of p62-driven BMAL1 degradation (Fig 3H–3J), suggesting that DRs are necessary for BMAL1 degradation. Furthermore, we assessed the regulatory role of p62 in the 20S proteasomal degradation of BMAL1 in *in vitro* analyses and discovered that p62 could promote BMAL1 degradation in a 20S proteasome reaction buffer (Fig 3K and 3L). These findings indicated that p62 destabilizes BMAL1 through ubiquitin-independent proteasomal degradation.

## Nuclear p62-driven BMAL1 degradation is not dependent on biomolecular condensates

Immunostaining studies indicated that p62 and BMAL1 accumulated in the nucleus during treated by LMB and MG132 (Fig 4A–4D). However, BMAL1 did not localize to p62-associated biomolecular condensates, where the proteasome is recruited to degrade substrates in a ubiquitin-dependent manner (Fig 4A and 4B). We also observed that co-expression with the p62 ΔNES (K7R) mutant, which eliminates p62's ability to facilitate condensate formation in the nucleus and inhibits condensate-associated protein degradation [7,8], did not increase the protein level of BMAL1 compared to co-expression with the p62 ΔNES mutant (Fig 4E–4G). This result is consistent with another p62 mutant, D69A, which abolishes PB1 domain-mediated p62 oligomerization, and similarly, the D69A mutation does not affect BMAL1 degradation driven by p62 (S3D Fig and Fig 3E). These results suggested that nuclear p62-mediated degradation of BMAL1 differs from that of p62-driven condensate- and ubiquitin-dependent proteasome degradation.

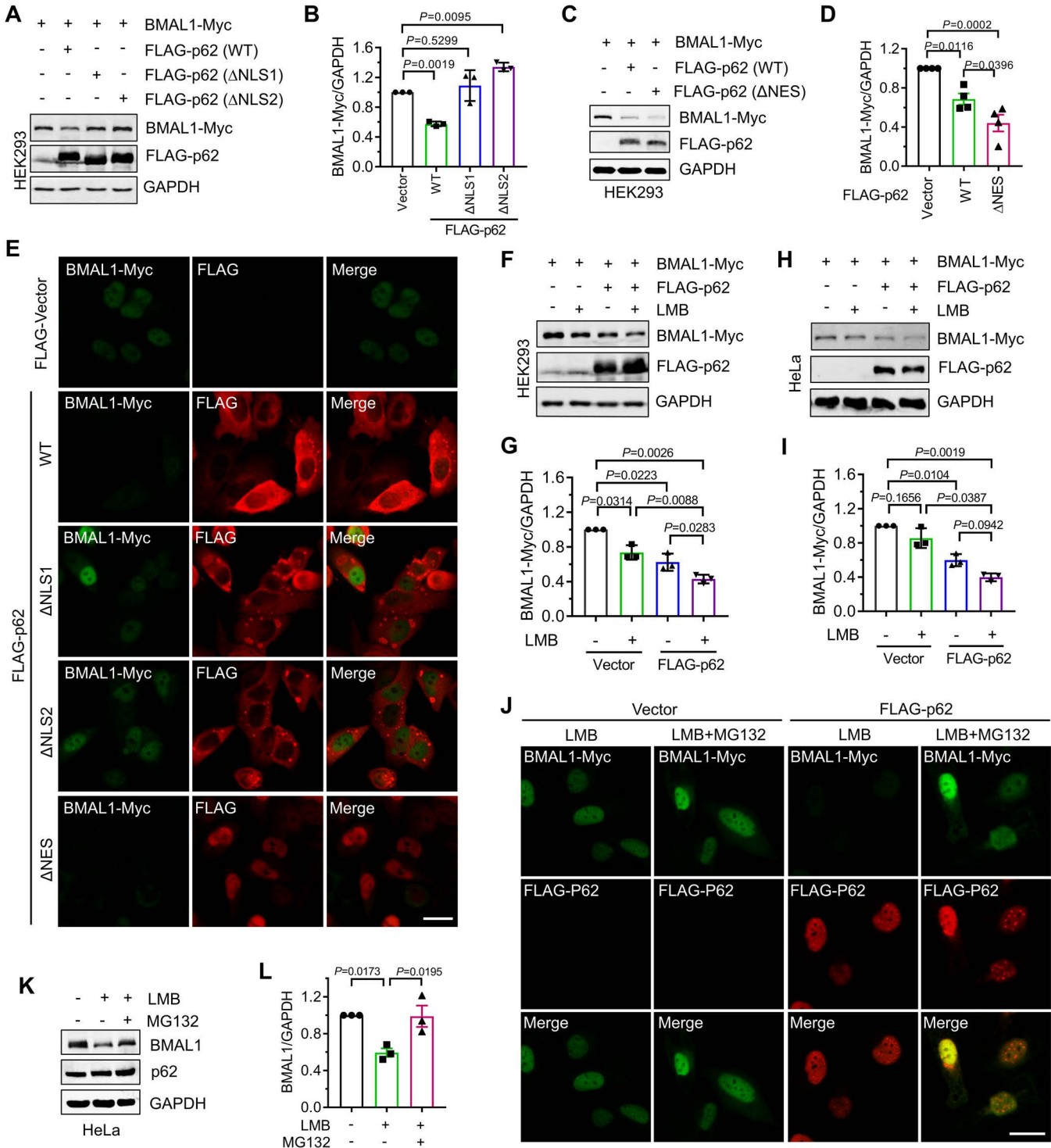

**Fig 2. p62 drives BMAL1 degradation in the nuclear.** (**A**) HEK293 cells were transfected with the indicated plasmids. After 24 h, the cells were lysed and subjected to western blot analysis with indicated antibodies. (**B**) Quantitative analysis of results in (**A**). Data are mean±SEM of biological replicates (n=3). (**C**) HEK293 cells were transfected with the indicated plasmids. After 24 h, the cells were lysed and subjected to western blot analysis with indicated antibodies. (**D**) Quantitative analysis of results in (**C**). Data are mean±SEM of biological replicates (n=3). (**E**) HeLa cells were transfected with BMAL-Myc and FLAG-p62 (WT or mutants). After 24 h, the cells were fixed and immunostained with indicated antibodies. Scale bar: 20 μm.

(**F**) HEK293 cells were transfected with the indicated plasmids. After 24 h, the cells were treated with Leptomycin B (LMB, 2 μM) for 16 h. The whole-cell lysates were subjected to western blot analysis with indicated antibodies. (**G**) Quantitative analysis of results in (**F**). Data are mean ± SEM of biological replicates ($n = 3$). (**H**) Hela cells were transfected with the indicated plasmids. After 24 h, the cells were treated with Leptomycin B (LMB, 2 μM) for 16 h. The whole-cell lysates were subjected to western blot analysis with indicated antibodies. (**I**) Quantitative analysis of results in (**H**). Data are mean ± SEM of biological replicates ($n = 3$). (**J**) HeLa cells were transfected with BMAL-Myc and FLAG-p62. After 24 h, the cells were treated with indicated drugs (LMB (2 μM), MG132 (2 μM)) for 16 h, then were fixed and immunostained with indicated antibodies. Scale bar: 20 μm. (**K**) Hela cells were treated with the indicated drugs (LMB (5 μM), MG132 (10 μM)) for 4 h. The whole-cell lysates were subjected to western blot analysis with indicated antibodies. (**L**) Quantitative analysis of results in (**K**). Data are mean ± SEM of biological replicates ($n = 3$). For **B**, **G**, and **I**, the P value was determined by Student's t test (two-sided). For **D** and **L**, the P value was determined by a one-way ANOVA analysis.

## p62 promotes the recruitment of BMAL1 to 20 S proteasome

In the nucleus, two distinct regulators, PA200 (PSME4) and PA28γ (REGγ or PSME3), have been reported to facilitate the 20S proteasome gate opening, thereby promoting substrate entry into the catalytic chamber and enabling degradation of non-ubiquitin-conjugated proteins [16,17]. However, our observations demonstrate that silencing of either PA200 or PA28γ does not affect the p62-induced reduction of BMAL1 (S4A–S4C Fig), suggesting that these regulators are not involved in the nuclear degradation of BMAL1 mediated by p62. Given that p62 is associated with proteasomes in the nucleus, we hypothesized that p62 functions as a proteasome receptor that recruits BMAL1 to the 20S proteasome. We first exzamined the interaction between p62 and 20S proteasome to investigate this hypothesis. We observed that nuclear p62 can co-immunoprecipitate 20S proteasome subunit α5 (Fig 5A). Moreover, *in vitro* binding analysis showed that TB domain of p62 was essential for the interaction of p62 to 20S proteasome (Fig 5B and 5C). Next, we examined the interaction between p62 and BMAL1. Our results demonstrated that nuclear p62 can co-immunoprecipitate BMAL1 (Fig 5D). Subsequently, we aimed to identify the interaction domains of both proteins using deletion mutants. Full-length FLAG-tagged p62 and various internal domain deletion mutants were co-expressed with Myc-tagged BMAL1 in HEK293 cells. Co-immunoprecipitation analysis using an anti-FLAG antibody revealed that the ZZ domain was essential for the binding of p62 to BMAL1 (Fig 5B and 5E). We also constructed nuclear-localized p62 mutants lacking either the ZZ or TB domain and found that the deletion of either domain suppressed nuclear p62-mediated BMAL1 degradation (S4D–S4F Fig). This finding further confirms the crucial role of these two domains in regulating BMAL1 protein stability. Besides, we co-expressed FLAG-p62 with either full-length Myc-tagged BMAL1 or its deletion mutants and analyzed their interactions. Only the BMAL1 variants containing the transactivation domain (TAD) interacted with p62 (Fig 5F and 5G), signifying that TAD is essential for this interaction.

We investigated whether p62 promotes the binding of BMAL1 to the proteasome. We transiently expressed Myc-tagged BMAL1 with or without HA-tagged p62 in HEK293 cells. Following cross-linking with dimethyl 3,3′-dithiobispropionimidate (DTBP), the cells were lysed and immunoprecipitated using Myc antibody. Co-immunoprecipitation studies demonstrated that co-expression of p62 enhanced the interaction between BMAL1 and 20S proteasome subunits (Fig 5H). Moreover, we transiently expressed Myc-tagged BMAL1 in wild-type (WT) and p62 knockout cells, treated with or without a proteasome inhibitor combined with LMB, to evaluate whether the deletion of p62 could block the binding of BMAL1 to the 20S proteasome. We discovered that LMB treatment significantly increased the interaction between BMAL1 and proteasome subunits (Fig 5I). Additionally, the absence of p62 significantly inhibited the association of BMAL1 with 20S proteasome subunits (Fig 5I). Since the ZZ and TB domains are critical for p62 binding to BMAL1 and the 20S proteasome, respectively, we examined the effect of deleting these domains on BMAL1 binding to the 20S proteasome. As expected, we found that the deletion of either the ZZ domain or the TB domain suppressed the interaction between BMAL1 and the 20S proteasome (S4G Fig). Interestingly, we also found that deleting DR2 and DR4 blocked the interaction between BMAL1 and the 20S proteasome (S4H Fig). This suggests that the IDRs may influence both the binding of the 20S proteasome to BMAL1 and its proteolysis. In summary, our findings revealed that p62 is a proteasome receptor that recruits BMAL1 to the 20S proteasome for degradation (Fig 5J).

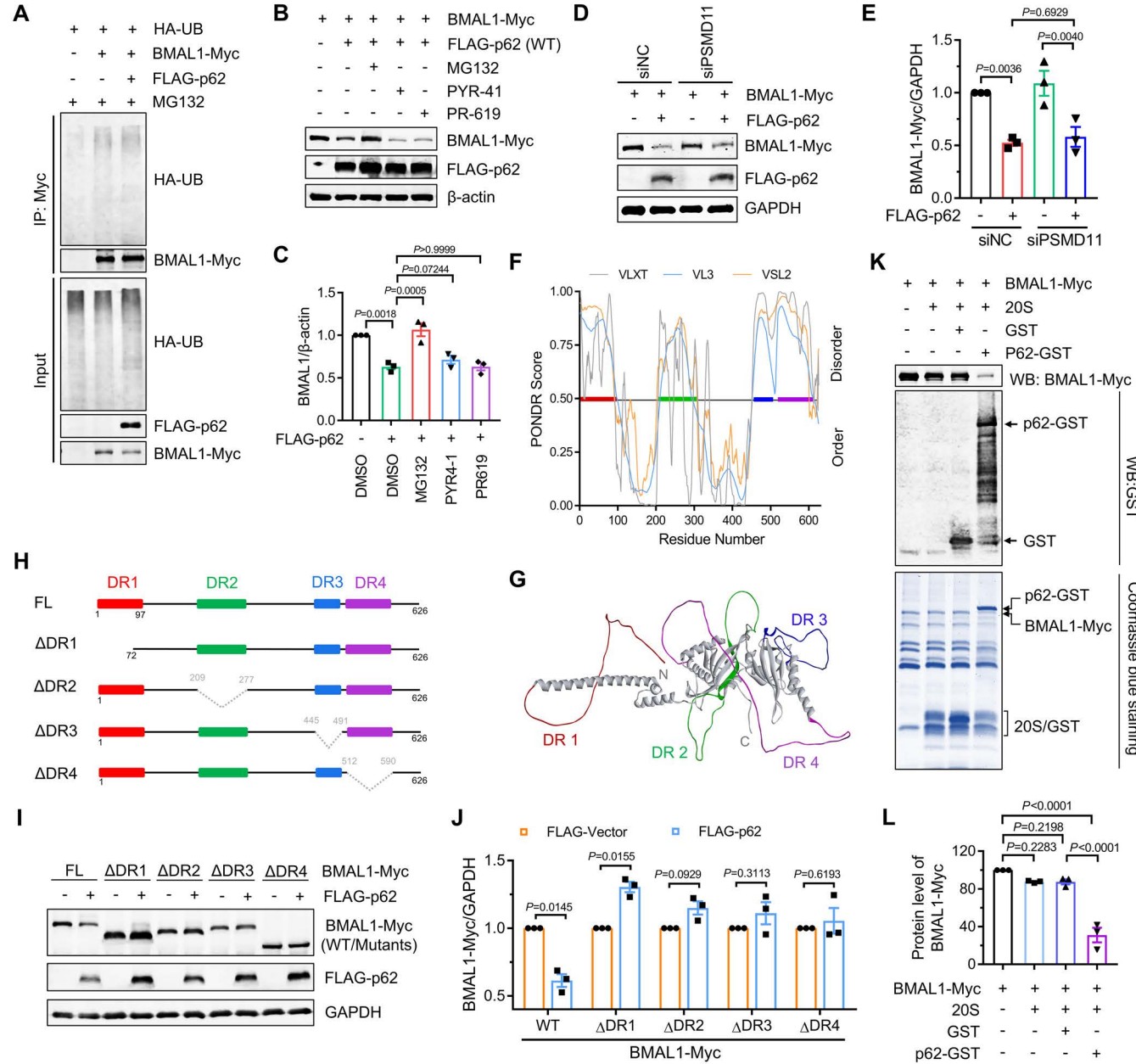

**Fig 3. p62 destabilizes BMAL1 through ubiquitin-independent proteasomal degradation.** (**A**) HEK293 cells that were transfected with the indicated plasmids. After 24 h, the cells were treated with MG132 (10 μM) for 5 h. Immunoprecipitation was then performed using anti-Myc antibody. The results were analysed by western blot with indicated antibodies. (**B**) HEK293 cells were transfected with the indicated plasmids. After 24 h, the cells were treated with the indicated drugs (MG132 (2 μM), PYR-41 (10 μM), PR-619 (10 μM)) for 16 h. The whole-cell lysates were subjected to western blot analysis with indicated antibodies. (**C**) Quantitative analysis of results in (**B**). Data are mean ± SEM of biological replicates (*n* = 3). (**D**) HEK293 cells were transfected with the indicated siRNAs. After 48 h, the cells were transfected with the indicated plasmids. After expressing 24 h, the cells were lysed and subjected to western blot analysis with indicated antibodies. (**E**) Quantitative analysis of results in (**D**). Data are mean ± SEM of biological replicates (*n* = 3). (**F**) Disorder prediction of the BMAL1 by PONDR (http://pondr.com). The y axis is the PONDR score and the x axis shows the residue number. Region that exceeds a score of 0.5 is considered to be disordered. (**G**) The disordered regions of BMAL1 shown in 3D structure (AlphaFold: AF-O00327-F1). (**H**) Structure of WT BMAL1 and its disordered region deletion mutants. (**I**) HEK293 cells were transfected with the indicated plasmids. After 24 h, the cells were lysed and subjected to western blot analysis with indicated antibodies. (**J**) Quantitative analysis of results in (**I**). Data are mean ± SEM of biological replicates (*n* = 3). (**K**) In vitro degradation assays containing BMAL1-Myc immunopurified from p62 knockout AD293 cells, 20S proteasome, and GST-p62

or GST. After reaction, the protein levels were detected by SDS-PAGE and Western blot analysis. **(L)** Quantitative analysis of the results in **(K)**. Data are mean±SEM of biological replicates ($n=3$). For **C**, **E**, and **J**, the $P$ value was determined by Student's t test (two-sided). For **L**, the $P$ value was determined by a one-way ANOVA analysis.

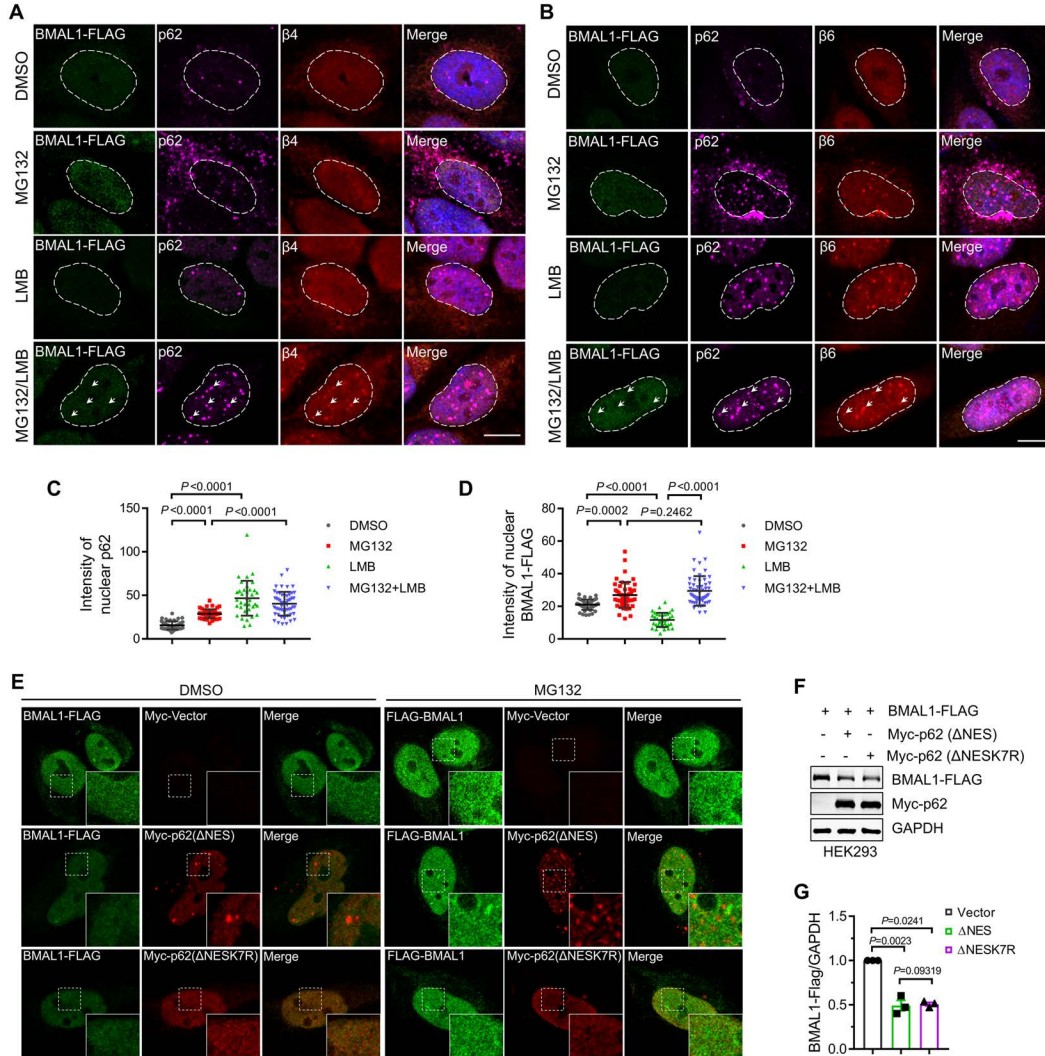

**Fig 4. Nuclear p62-driven BMAL1 degradation is dot dependent on biomolecular condensates. (A-B)** HeLa cells were transfected with BMAL1-FLAG and then treated with the indicated drugs (LMB (2 μM), MG132 (2 μM)) for 16 h. the cell were fixed and immunostained with indicated antibodies. Nuclei were stained with DAPI (blue). Scale bar: 10 μm. **(C-D)** Quantitative analysis of the results in (A and B). Data are mean±SEM from three biological replicates and each dot represents one cell ($n=36-55$). **(E)** HeLa cells were transfected with BMAL1-FLAG and mutated p62 and then treated with MG132 (2 μM) or not for 16 h. The cells were fixed and immunostained with indicated antibodies. Scale bar: 10 μm. **(F)** HEK293 cells were transfected with the indicated plasmids. After 24 h, the cells were lysed and subjected to western blot analysis with indicated antibodies. **(G)** Quantitative analysis of the results in **(F)**. Data are mean±SEM of biological replicates ($n=3$). For **C**, **D**, and **G**, the $P$ value was determined by a one-way ANOVA analysis.

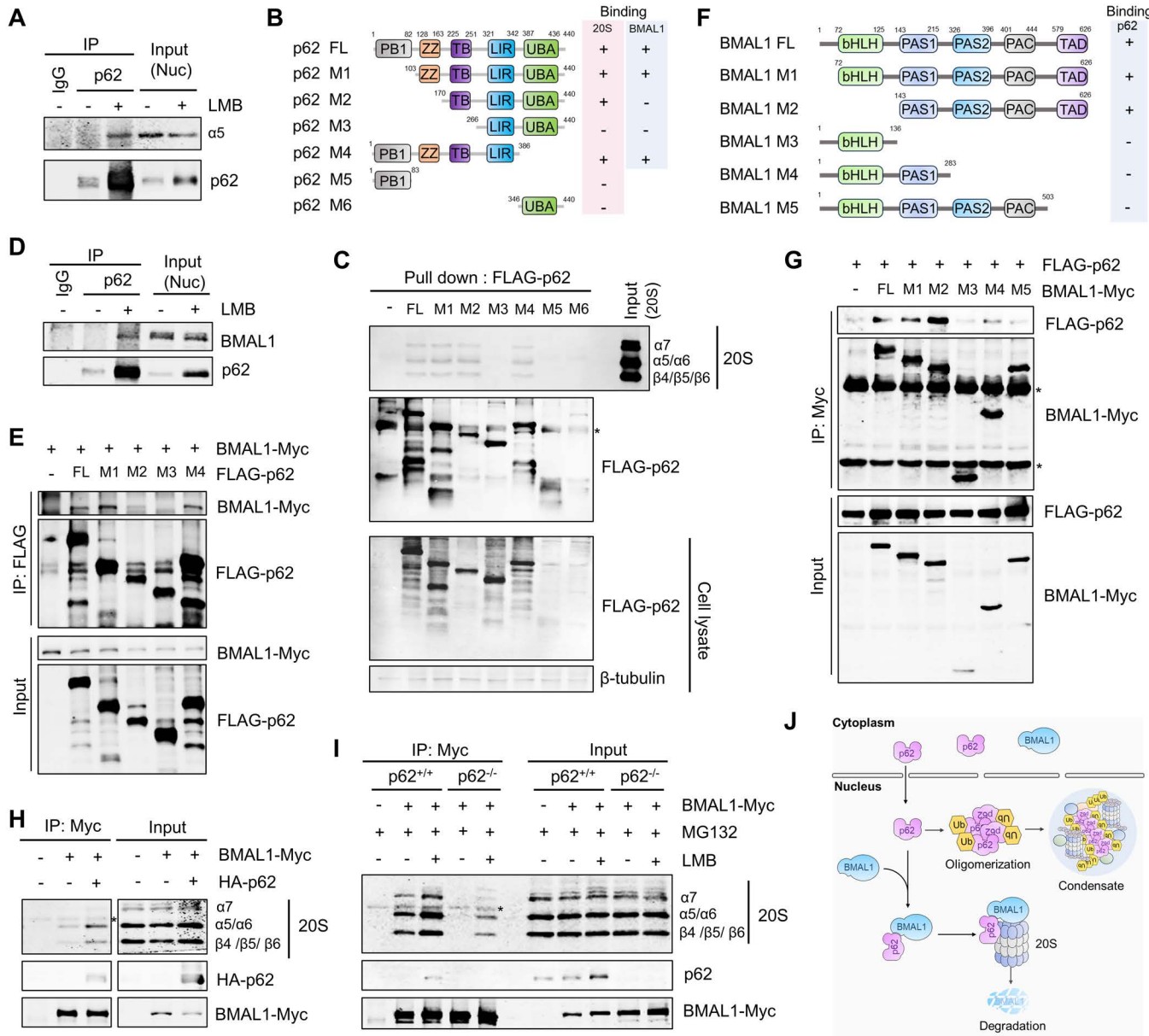

**Fig 5. p62 promotes the recruitment of BMAL1 to 20 S proteasome.** (**A**) HEK293 cells were treated with/without LMB (5 μM) for 16 h, the nuclear fractions were used to perform immunoprecipitation using anti-p62 antibody. The results were analyzed by western blot with the indicated antibodies. (**B**) Illustration of p62 domain organization and respective mutants. (**C**) p62⁻/⁻ cells were transfected with indicated plasmids that expressing FLAG-tagged WT or mutated p62. After 48 h, the expressed p62 were immunopurified using anti-FALG magnetic beads. The magnetic beads containing p62 (WT or mutants) were then incubated with 20S proteasome in the presence of 50 μM MG132. The beads were washed and subjected to western blot analysis using the indicated antibodies. (**D**) HEK293 cells were treated with/without LMB (5 μM) for 16 h, the nuclear fractions were used to perform immuno-precipitation using anti-p62 antibody. The results were analyzed by western blot with the indicated antibodies. (**E**) p62⁻/⁻ cells were transfected with the indicated plasmids. After 24 h, the cells were treated with MG132 (10 μM) for 5 h and then used to immunoprecipitation with anti-FLAG antibody. The results were analyzed by western blot with the indicated antibodies. (**F**) Illustration of BMAL1 domain organization and respective mutants. (**G**) HEK293 cells were transfected with the indicated plasmids. After 24 h, the cells were treated with MG132 (10 μM) for 5 h and then used to immunoprecipitation with anti-Myc antibody. The results were analyzed by western blot with the indicated antibodies. (**H**) HEK293 cells were transfected with the indicated plasmids. After 24 h, the cells were cross-linked with DTBP, and then used to immunoprecipitation with anti-Myc antibody. The results were analyzed by western blot with the indicated antibodies. (**I**) p62⁺/⁺ and p62⁻/⁻ cells were transfected with BMAL1-Myc plasmids, and then treated with the indicated drugs (MG132 (2 μM), LMB (2 μM)) for 16 h. After cross-linked with DTBP, the cells were lysed and used to perform immunoprecipitation using anti-Myc

antibody. The results were analyzed by western blot with the indicated antibodies. (**J**) Model depicting nuclear p62 promoting the recruitment of BMAL1 to the 20S proteasome and facilitating its ubiquitin-independent proteasomal degradation. α5, 20S proteasome subunit α5; α6, 20S proteasome subunit α6; α7, 20S proteasome subunit α7; β4, 20S proteasome subunit β4; β5, 20S proteasome subunit β5; β6, 20S proteasome subunit β6.

## Targeting nuclear p62 suppresses BMAL1-associated tumor cell growth

BMAL1 has been identified as a crucial regulator of tumorigenesis [10]. Consequently, we investigated whether re-modulating the nuclear accumulation of p62 could inhibit tumor cell growth mediated by BMAL1. Initially, we examined the relationship between p62 and BMAL1 protein levels in samples from patients with colon cancer. We observed a significant increase in BMAL1 protein levels in colon cancer tissues compared to adjacent tissues (Fig 6A and 6B). Similarly, p62 protein levels were also significantly elevated in colon cancer tissues, and its expression showed a positive correlation with BMAL1 expression (S5A–S5C Fig). However, in colon cancer cell lines (RKO and SW480), overexpression of p62 did not result in an increase in BMAL1 protein levels (Fig 6C–6F). Interestingly, the expression of exogenous p62 ΔNES resulted in substantial downregulation of BMAL1 protein levels and inhibited tumor cell growth both in RKO and SW480 cells, whereas WT p62 expression did not provide a comparable impact (Fig 6C–6L). Furthermore, we observed that p62 ΔNES-mediated inhibition of colon cancer cell growth could be diminished by BMAL1 knockdown (Fig 6G–6L), suggesting that p62 ΔNES suppresses colon cancer cell growth by reducing BMAL1 protein levels. Additionally, we observed that BMAL1 also did not colocalize with nuclear p62-associated condensates in colon cancer cells (S5D Fig). Moreover, we found that deleting ZZ or TB domain in p62 blocked the inhibitory effect of nuclear p62 on colon cancer cell growth (S5E–S5H Fig), suggesting that nuclear p62-mediated BMAL1 degradation is critical for the tumor growth inhibition. In summary, these results indicate that remodeling the nuclear accumulation of p62 may represent a potential strategy for targeting BMAL1 to suppress tumor cell growth.

## Discussion

A crucial function of p62 is its role as an autophagy receptor that mediates the autophagic degradation of proteins. Recently, researchers have been focusing on the significance of p62 in mediating protein degradation within the nucleus. Fu et al. demonstrated that nuclear p62 promotes protein degradation through the ubiquitin-dependent proteasomal pathway by forming biomolecular condensates [7,8]. Our study elucidates a novel mechanism by which nuclear p62 facilitates proteasomal degradation of the transcription factor BMAL1 in a manner that is independent of both ubiquitin and condensate formation, highlighting p62 involvement in nuclear protein quality control.

Although p62 is a well-known autophagy cargo receptor, it also functions as a proteasome cargo receptor. Studies have reported that the PB1 domain of p62 interacts with the subunits of the 19S proteasome, including Rpn10 and Rpn1, facilitating the shuttling of polyubiquitinated substrates bound to its UBA domain to the proteasome for degradation [18,19]. Recent studies have demonstrated that nuclear p62 drives the degradation of nuclear proteins through the UPS [7,8]. This function of p62 is consistently associated with its role in driving biomolecular condensates, which concentrate components of the UPS machinery, including proteasomes and ubiquitin ligases [7,8]. Although the mechanisms by which p62 organizes these condensates and delivers substrates for degradation remain unclear, research indicates that oligomerization mediated by the PB1 domain and the binding of its UBA domain to ubiquitin conjugates are critical for the formation of these condensates and the subsequent degradation of nuclear proteins [7]. Our findings indicate that nuclear p62 promotes the proteasomal degradation of BMAL1 in a ubiquitin-independent manner. It was observed that BMAL1 did not enter the p62-driven condensate. Besides, mutants that abolished the p62 condensate function (ΔNESK7R) significantly promoted BMAL1 degradation, similar to the ΔNES mutant. These findings indicate that p62 drives the formation of condensates to enhance ubiquitin-dependent proteasomal degradation of nuclear proteins and facilitates proteasomal degradation of nuclear proteins through a condensate- and ubiquitin-independent mechanism. Interestingly, we found that the

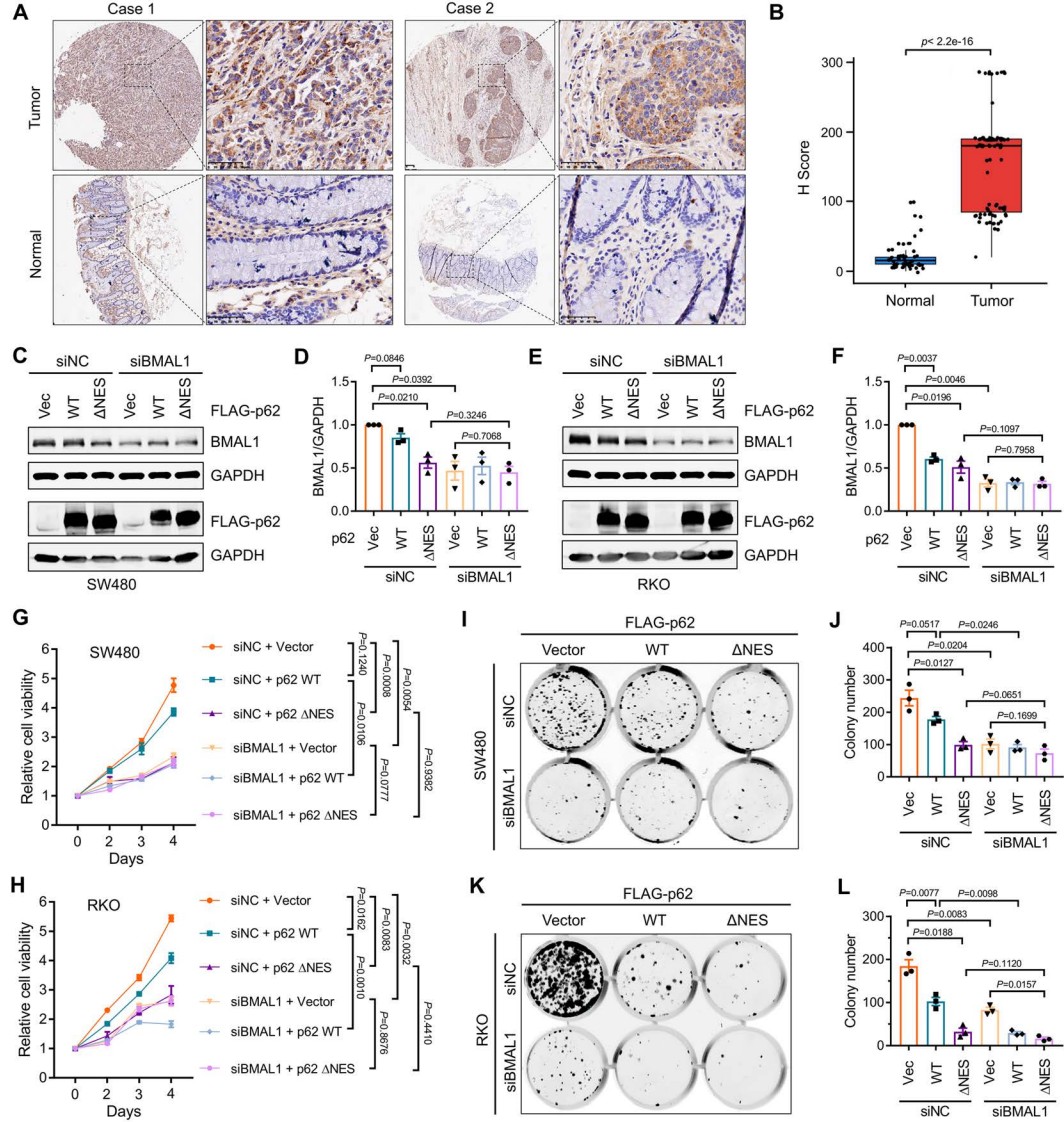

**Fig 6. Targeting nuclear p62 suppresses BMAL1-associated tumor cell growth. (A)** Representative images of BMAL1 expression in tissue microarray comprising colon cancer tissues and paired normal adjacent tissues. Scale bar: 50 µm. **(B)** The histoscore scores (H Score) of BMAL1 in tumor (*n* = 86) and normal adjacent tissues (*n* = 53). **(C)** SW480 cells were transfected with the indicated siRNA and plasmids. After 72 h, the cells were lysed and subjected to western blot analysis with indicated antibodies. **(D)** Quantitative analysis of results in **(C)**. Data are mean ± SEM of biological replicates (*n* = 3). **(E)** RKO cells were transfected with the indicated siRNA and plasmids. After 72 h, the cells were lysed and subjected to western blot analysis with indicated antibodies. **(F)** Quantitative analysis of results in **(E)**. Data are mean ± SEM of biological replicates (*n* = 3). **(G-H)** SW480 (**G**) and RKO (**H**) were transfected with the indicated siRNA and plasmids. The cell viability at various times was examined by CCK-8 analysis. Data are mean ± SEM of biological replicates (*n* = 3). **(I)** Colony formation of SW480 cells transfected with indicated plasmids and siRNA. **(J)** Quantitative analysis of results in (I). Data are mean ± SEM of biological replicates (*n* = 3). **(K)** Colony formation of RKO cells transfected with indicated plasmids and siRNA. **(L)** Quantitative analysis of results in **(K)**. Data are mean ± SEM of biological replicates (*n* = 3). For **B**, the *P* value was determined by Wilcoxon rank-sum test. For **D**, **F**, **G**, **H**, **J**, and **L**, the *P* value was determined by Student's t test (two-sided).

PB1 and UBA domains of p62 are crucial for mediating BMAL1 degradation, but this does not rely on their role in condensate formation. This suggests that the PB1 and UBA domains may regulate substrate degradation by the 20S proteasome through an unknown molecular mechanism, which warrants further investigation in future studies. Many IDPs are prone to

degradation by the 20S proteasome without requiring ATP consumption or ubiquitination [3]. Our study's PONDR analysis revealed that BMAL1 contains multiple IDRs crucial for p62-mediated BMAL1 degradation. We also discovered that the recognized proteasomal activators PA28γ and PA200 were not implicated in this process. These results suggested that p62 may be a novel proteasomal activator that acts alone or cooperates with other unknown molecules to open the 20S proteasome gate. This mechanism warrants further investigation in future studies.

Tang et al. demonstrated that the ferroptosis inducer RSL3 can induce the autophagic degradation of BMAL1, which facilitates cellular ferroptosis [20]. They discovered that p62 was responsible for delivering BMAL1 to the autophagosome [20]. Our study revealed that nuclear p62 can similarly deliver BMAL1 to the 20S proteasome for degradation. These findings suggested that p62 is an essential regulatory factor in maintaining BMAL1 protein stability, although the sorting mechanisms underlying the p62-mediated degradation of BMAL1 remain unclear. The distinct degradation pathways of BMAL1 mediated by p62 may depend on the upstream regulatory signals. Unlike other ferroptosis inducers, Tang et al. observed that only RSL3 could trigger p62-mediated autophagic degradation of BMAL1 [20]. We further demonstrated that only nuclear p62 could drive proteasomal degradation of BMAL1. Another potential factor influencing BMAL1 degradation is cellular distribution. The RSL3-induced interaction between BMAL1 and p62 depends on the UBA domain of p62, suggesting that the ubiquitination of BMAL1 may be a key modification for driving its autophagic degradation. Previous studies have identified several E3 ligases that catalyze the ubiquitination of BMAL1 and promote its proteasomal degradation [21–23]. These findings imply that cytosolic BMAL1 may undergo ubiquitination, leading to its degradation through either p62-mediated autophagy or UPS. A critical inquiry for forthcoming research is elucidating the molecular mechanisms sorting BMAL1 for degradation through these distinct pathways.

BMAL1 is a crucial protein involved in regulating circadian rhythm. Disarrangement of circadian rhythms is involved in developing various diseases [10–13]. In this study, we observed that p62 deletion increased the expression of most BMAL1-targeted circadian genes and reduced the transcription of *BMAL1* and *CLOCK*, suggesting that nuclear p62 may regulate circadian rhythms by modulating the stability of BMAL1. In future research, investigating the effects of p62 knockout or nuclear translocation loss on the animal circadian clock will clarify the regulatory role of p62 in circadian rhythms. Accordingly, targeting p62 could represent a potential therapeutic strategy for circadian rhythm-associated diseases, which warrants further attention in future research. Recently, the significance of BMAL1 in tumorigenesis and cancer therapy has garnered heightened attention [10,24,25]. The role of BMAL1 may vary across different tumor types. In several cancers, including HCC [26,27], pancreatic cancer [28,29], and tongue squamous cell carcinoma [30], low expression of BMAL1 has been associated with poor prognosis, suggesting that BMAL1 functions as a tumor suppressor and that increasing its expression may be a potential anti-cancer strategy. However, in colon cancer [31] and breast cancer [32], BMAL1 appears to play an oncogenic function. Interestingly, here, we found the BMAL1 protein levels was positively correlated with that of p62 in colon cancer tissues. However, overexpression of wild-type p62 in colon cancer cells did not increase BMAL1 protein levels, suggesting p62 may regulate BMAL1 expression through multiple pathways, which could lead to opposing effects on BMAL1 expression. One possible explanation is that the retention efficiency of p62 in the nucleus of colon cancer cells is insufficient to promote the degradation of BMAL1. Consistently, we found that overexpression of nuclear-localized p62 enhanced the degradation of endogenous BMAL1 and suppressed tumor cell growth. These results indicated that targeting nuclear p62, rather than cytoplasmic p62, may suppress BMAL1-associated tumor cell growth. Future research must concentrate on developing drugs that can accurately regulate the nuclear-cytoplasmic distribution of p62, which may alter the protein levels of BMAL1 in cells and be applied in cancer therapy.

In summary, this study revealed that nuclear p62 can promote the recruitment of BMAL1 to the proteasome, thereby facilitating its ubiquitin-independent proteasomal degradation. Our preliminary findings also suggested that remodeling the accumulation capacity of p62 in the nucleus may represent a potential strategy for targeting BMAL1 in cancer therapy.

## Materials and methods

### Ethics statement

The human colon cancer tissue microarray was provided by the Ethics Committee of Taizhou Hospital of Zhejiang Province. All the patients signed informed consent forms.

### Cell culture, transfection, viability, and colony formation assay

HEK293, Hela, SW480, and RKO cells were obtained from ATCC. Wild-type AD293 (p62$^{+/+}$) and p62 knockout AD293 (p62$^{-/-}$) were obtained and constructed as described in previous studies [33]. All cells were cultured in Dulbecco's modified Eagle's medium containing 10% fetal bovine serum (FBS, HyClone, SH30071.03), 1% penicillin–streptomycin (Thermo Fisher Scientific, 15140211), 2 mM l-glutamine (Thermo Fisher Scientific, 25030081) and 1 × non-essential amino acids (NEAAs, Thermo Fisher Scientific, 11140076) at 37°C in a 5% CO2 incubator. All the cell lines were authenticated and tested for contamination. siRNA transfections were carried out with Lipofectamine 2000. Plasmid transfection was carried out with Megatran (OriGene, TT200003) for Hela and HEK293 cells, while Lipofectamine 2000 (Thermo Fisher Scientific, 11668019) was utilized for other cells. All transfections were performed according to the manufacturer's instructions. For the cell viability assay, cells were seeded into 96-well plates. After transfected with indicated plasmids or siRNA, cell viability was measured using Cell Counting Kit-8 (TargetMol, C0005) following the protocols. For colony formation assay, cells were seeded into 12-well plates and transfected with indicated plasmids or siRNA. After 1 week, cells were fixed with methanol and then stained with 0.1% crystal violet. Following washed with PBS, the picture of the plates was obtained by the Li-Cor Odyssey Clx Infrared Imaging System, and the numbers of colonies were counted manually.

### Plasmid construction and siRNA

FLAG-ubiquitin (UB) and FLAG-, HA- or Myc-tagged p62 have been described in previous studies [34,35]. To construct Flag- or Myc-tagged wild-type or mutated BMAL1 and mutated p62, the DNA fragments were obtained by PCR and then cloned into the pcDNA3.1 vector with indicated tags through Uniclone One Step Seamless Cloning Kit (GeneSand, Beijing). S1 Table lists the primer sequence information used for DNA fragment amplification. DNA sequencing was utilized to validate all the plasmids. siRNAs corresponding indicated genes were purchased from Tsingke (Chengdu, China). S2 Table lists the siRNA sequences.

### Chemical reagents

MG132 (MCE, HY-13259), Bafilomycin A1 (MCE, HY-100558), Pepstatin A (MCE, HY-P0018), E64d (MCE, HY-100229), Leptomycin B (MCE, HY-16909), PYR-41 (MCE, HY-13296), PR-619 (MCE, HY-13814), DTBP (Thermo Fisher Scientific, 20665), DAPI (Beyotime, C1006).

### Protein extraction

Total protein extract was acquired by homogenizing the cells in 1 × SDS sample buffer. The nuclear and cytoplasmic fractions were separated using the following protocol. After harvesting, cells ($5 \times 10^6$) were washed twice with cold PBS, resuspended in 500 μL hypotonic buffer (10 mM NaCl, 3 mM MgCl$_2$, 20 mM Tris-HCl, pH 7.4), and protease inhibitor mixture (Thermo Fisher Scientific, 87785)), and incubated on ice for 15 min. Then, 25 μL detergent (10% NP-40) was added and vortexed vigorously for 10 s. The lysates were centrifuged for 10 min at $200 \times g$ at 4°C. The supernatant (cytoplasmic fraction) was collected, and the pellet (nuclear fraction) was washed with hypotonic buffer. The pellet was then homogenized with 1 × SDS sample buffer for SDS-PAGE or lysed with NP-40 Alternative cell lysis buffer (150 mM NaCl, 1mM EDTA, 1% NP-40 Alternative (Merck Millipore, 492016), 50 mM Tris-HCl (pH 8.0), and protease inhibitor mixture (Thermo Fisher Scientific, 87785)) following ultrasonication for immunoprecipitation.

## Immunoprecipitation

Cells or nuclear fractions were incubated with NP-40 alternative cell lysis buffer containing protease inhibitor mixtures (Thermo Fisher Scientific, 87785), sonicated three times or 5 s at 35% amplitude, and then lysed for 30 min on ice. For co-immunoprecipitation of proteasome subunits, cells were cross-linked with 0.3 µg/mL DTBP (Thermo Fisher Scientific, 20665) in PBS for 30 min at room temperature before lysis. The lysates were centrifuged for 15 min at 14,000 × $g$ at 4°C, and the supernatant was collected. Subsequently, primary antibodies were added and incubated for 2 h or overnight while rotating at 4°C. The lysates were incubated with protein-A/G agarose beads (MCE, HY-K0230) for 1 h at 4°C, washed thrice with NP-40 alternative cell lysis buffer, and resuspended in 1 × SDS sample buffer. After boiling for 5 min, samples were subjected to Western blotting analysis.

## Western blotting analysis

In brief, protein samples were separated by SDS-PAGE and transferred to an Immunobilon-FL PVDF membrane (Merck Millipore, IPFL00010). The membrane was blocked with 5% BSA in TBST and incubated with primary antibodies overnight at 4°C. After washing thrice with TBST, the membrane was incubated with secondary antibodies for 2 h at room temperature in the dark. After washing thrice with TBST, the membrane was analyzed using a Li-Cor Odyssey Clx Infrared Imaging System (LI-COR Biotechnology, Lincoln, NE, USA). Quantitative analysis was conducted using ImageJ software. Information regarding the antibodies used is listed in S3 Table.

## Immunostaining

Immunostaining was performed as described previously [36]. Briefly, after treatment, cells were fixed with 4% paraformaldehyde for 15 min and permeabilized with 0.2% Triton X-100 (Merck Millipore, 94101-L) for 15 min. After blocking with 5% goat serum for 1 h, cells were incubated with primary antibodies overnight at 4°C or for 2 h at room temperature. The cells were washed and incubated with the appropriate secondary antibodies for 1 h at room temperature. Nuclei were stained with a DAPI solution. Images were captured using a fluorescence microscope (Nikon Eclipse 80i equipped with Nikon PLAN FLUOR ×40 objective) or a Leica Stellaris 5 confocal microscope. Photographic images were resized and analyzed using the ImageJ software. Information regarding the antibodies used is listed in S3 Table.

## Real-time RT-qPCR

Following the protocol, total cellular RNA was extracted with the cell total RNA isolation kit (ForeGene, RE-03111). Subsequently, complementary DNAs (cDNAs) were synthesized using the Prime Script 1st strand cDNA Synthesis Kit (TaKaRa, 6110A). cDNAs were analyzed by qPCR using SYBR Green (Selleck, B21202) and gene-specific primers. S4 Table lists the PCR primer sequences.

## *In vitro* degradation assay

AD293 p62$^{-/-}$ cells were cultured in two 10-cm dishes and transfected with BMAL1-Myc. After 24 h, the cells were lysed with NP-40 alternative cell lysis buffer containing protease inhibitor mixtures (Thermo Fisher Scientific, 87785) on ice for 30 min and then centrifuged at 14,000 × $g$ for 15 min at 4°C. For the supernatant, BMAL1-Myc was immunopurified using anti-Myc Magnetic Beads (MCE, HY-K0206). Then, the beads were washed thrice with NP-40 alternative cell lysis buffer and once with the reaction buffer (50 mM Tris-HCl (pH 7.5), 20 mM NaCl, 1 mM EDTA, 2 mM MgCl$_2$, 2 mM DTT, and 5% glycerol). Purified BMAL1 was then incubated with 1 µg of 20S proteasome (Enzo Life Sciences, BML-PW8720-0050) in the presence of 1 µg of GST-p62 (Proteintech, Ag13131) or GST (Proteintech, Ag0040) at 37°C for 6 h. The reaction was stopped with an SDS sample buffer, and digestion products were analyzed using SDS-PAGE following Western blotting.

### *In vitro* binding assay of p62 and 20S proteasome

AD293 p62$^{-/-}$ cells were cultured in 10-cm dishes and transfected with WT or mutated FLAG-p62. After 24 h, the cells were lysed with NP-40 alternative cell lysis, and FLAG-p62 (WT or mutants) was immunopurified with anti-FLAG magnetic beads (MCE, HY-K0207). Then, the beads were washed thrice with NP-40 alternative cell lysis buffer and resuspended in 500 μL NP-40 alternative cell lysis buffer containing 50 μM MG132 and 2 μg of 20S proteasome. After incubation for 2 h at 4°C, the beads were washed thrice with NP-40 alternative cell lysis buffer and resuspended in 1 × SDS sample buffer for Western blotting analysis.

### Tissue microarray and immunohistochemistry

The human colon cancer tissue microarray prepared by Shanghai Outdo Biotech, China. All the patients signed informed consent forms. This study was approved by the Ethics Committee of Taizhou Hospital of Zhejiang Province. Immunostaining was performed using against human BMAL1 (Proteintech, 14268–1-AP, 1: 100) and p62 (Proteintech, 18420-1-AP, 1: 200) antibodies. The histoscore (H Score) was calculated using the formula: H Score= (percentage of positive cells (ranging from 0% to 100%) × the average intensity of the positive staining (1+, 2+, or 3+)).

### Statistical analysis

Statistical analyses were performed using GraphPad Prism 7.0. Data are presented as mean ± standard error of the mean (SEM) of three or more biological replicates. Comparisons between individual data points were conducted as described in the figure legends. A $P < 0.05$ was considered statistically significant.

### Supporting information

**S1 Fig. p62 regulates the expression of clock genes by regulating BMAL1 protein levels, related to** Fig 1. p62$^{+/+}$ and p62$^{-/-}$ cells were transfected with indicated siRNAs. After 48 h, the cells were treated with Dexamethasone (100 μM) for 2 h. The mRNA levels of clock genes were then analysis at 4 h (**A**) and 16 h (**B**). Data are mean ± SEM of biological replicates ($n = 3$). The $P$ value was determined by a one-way ANOVA analysis.
(TIF)

**S2 Fig. p62 drives BMAL1 degradation in the nuclear, related to** Fig 2. (**A**) p62 knock out AD293 (p62$^{-/-}$) cells were transfected with the indicated plasmids. After 24 h, the cells were lysed and subjected to western blot analysis with indicated antibodies. (**B**) Quantitative analysis of results in (**A**). Data are mean ± SEM of biological replicates ($n = 3$). (**C**) HEK293 cells were treated with LMB (2 μM)) for 16 h. The whole-cell lysates were subjected to western blot analysis with indicated antibodies. (**D**) Quantitative analysis of results in (**C**). Data are mean ± SEM of biological replicates ($n = 3$). For **B** and **D**, the $P$ value was determined by Student's t test (two-sided).
(TIF)

**S3 Fig. p62 destabilizes BMAL1 through ubiquitin-independent proteasomal degradation, related to** Fig 3. (**A-C**) HeLa cells were transfected with FLAG-ubiquitin (FLAG-UB) and BMAL1-Myc, then treated with MG132 (2 μM) and LMB (2 μM) for 16 hours. Cells were subsequently immunostained with the indicated antibodies. Nuclei were stained with DAPI (blue). Scale bar: 10 μm. (**D**) p62 knock out AD293 (p62$^{-/-}$) cells were transfected with the indicated plasmids. After 24 h, the cells were lysed and subjected to western blot analysis with indicated antibodies. (**E**) Quantitative analysis of results in (**D**). Data are mean ± SEM of biological replicates ($n = 3$). (**F**) HEK293 cells were transfected with indicated siRNA for 48 h, the mRNA levels of *PSMD11* were analyzed by RT-qPCR. Data are mean ± SEM of biological replicates ($n = 3$). For **E** and **F**, the $P$ value was determined by Student's t test (two-sided).
(TIF)

**S4 Fig. p62 promotes the recruitment of BMAL1 to 20 S proteasome, related to Fig 5.** (**A**) HEK293 cells were transfected with indicated siRNA for 48 h, the mRNA levels of *PSME3* and *PSME4* were analyzed by RT-qPCR. Data are mean ± SEM of biological replicates (*n* = 3). (**B**) HEK293 cells transfected with indicated plasmids and siRNA for 72 h. The cells were then lysed and subjected to western blot analysis with indicated antibodies. (**C**) Quantitative analysis of results in (**B**). Data are mean ± SEM of biological replicates (*n* = 3). (**D**) Hela cells were transfected with the indicated plasmids that expressing FLAG-tagged p62 mutants. After 24 h, the cells were fixed and immunostained with anti-FLAG antibody. Nuclei were stained with DAPI (blue). Scale bar: 10 μm. (**E**) p62$^{-/-}$ cells were transfected with the indicated plasmids. After 24 h, the cells were lysed and subjected to western blot analysis with indicated antibodies. (**F**) Quantitative analysis of results in (**E**). Data are mean ± SEM of biological replicates (*n* = 3). (**G**) p62$^{-/-}$ cells were transfected with the indicated plasmids. After 24 h, the cells were treated with MG132 (2 μM) and LMB (2 μM) for 16 hours. After cross-linked with DTBP, the cells were lysed and used to perform immunoprecipitation using anti-Myc antibody. The results were analyzed by western blot with the indicated antibodies. (**H**) HEK293 cells were transfected with the indicated plasmids. After 24 h, the cells were cross-linked with DTBP, lysed and used to perform immunoprecipitation using anti-Myc antibody. The results were analyzed by western blot with the indicated antibodies. For **A** and **C**, the *P* value was determined by Student's t test (two-sided). For **F**, the *P* value was determined by a one-way ANOVA analysis.
(TIF)

**S5 Fig. Targeting nuclear p62 suppresses BMAL1-associated tumor cell growth, related to Fig 6.** (**A**) Representative images of p62 expression in a tissue microarray comprising colon cancer tissues and paired normal adjacent tissues. Scale bar: 50 μm. (**B**) H-score values for p62 in tumor (*n* = 86) and normal adjacent tissues (*n* = 53). the *P* value was determined by Wilcoxon rank-sum test. (**C**) Spearman correlation between BMAL1 and p62 H-scores values for tumor samples (*n* = 86). (**D**) SW480 cells were treated with MG132 (2 μM) and LMB (2 μM) for 16 h. the cells were then fixed and immunostained with indicated antibodies. Nuclei were stained with DAPI (blue). Scale bar: 10 μm. (**E**) SW480 cells were transfected with indicated plasmids expressing mutant p62. After 48 h, the cells were lysed and subjected to western blot analysis with indicated antibodies. (**F**) SW480 cells were transfected with indicated plasmids expressing mutant p62. After 72 h, the cell viability was examined by CCK-8 analysis. Data are mean ± SEM of biological replicates (*n* = 3). (**G**) RKO cells were transfected with indicated plasmids expressing mutant p62. After 48 h, the cells were lysed and subjected to western blot analysis with indicated antibodies. (**H**) RKO cells were transfected with indicated plasmids expressing mutant p62. After 72 h, the cell viability was examined by CCK-8 analysis. Data are mean ± SEM of biological replicates (*n* = 3). For **F** and **H**, the *P* value was determined by Student's t test (two-sided).
(TIF)

**S1 Table. Sequences of the primers used for ORF amplification.**
(XLSX)

**S2 Table. Sequences of siRNA.**
(XLSX)

**S3 Table. Antibody information.**
(XLSX)

**S4 Table. Sequences of qRT-PCR Primer.**
(XLSX)

**S1 Data. Source data file.**
(XLSX)

## Author contributions

**Conceptualization:** Chenliang Zhang.

**Data curation:** Chenliang Zhang.

**Formal analysis:** Liping Li.

**Funding acquisition:** Chenliang Zhang.

**Investigation:** Chenliang Zhang, Quanyou Wu, Huan Zhang, Ruichen Liu.

**Methodology:** Quanyou Wu, Huan Zhang.

**Project administration:** Chenliang Zhang.

**Software:** Quanyou Wu, Liping Li.

**Supervision:** Chenliang Zhang.

**Writing – original draft:** Chenliang Zhang, Quanyou Wu, Huan Zhang, Ruichen Liu, Liping Li.

**Writing – review & editing:** Chenliang Zhang.

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
