## [Decision Letter · Decision Letter 0]

PGENETICS-D-25-00276

Nuclear p62/SQSTM1 Facilitates Ubiquitin-independent Proteasomal Degradation of BMAL1

PLOS Genetics

Dear Dr. Zhang,

Thank you for submitting your manuscript to PLOS Genetics. After careful consideration, we feel that it has merit but does not fully meet PLOS Genetics's publication criteria as it currently stands. Therefore, we invite you to submit a revised version of the manuscript that addresses the points raised during the review process.

Please submit your revised manuscript within 60 days Jun 28 2025 11:59PM. If you will need more time than this to complete your revisions, please reply to this message or contact the journal office at plosgenetics@plos.org. Please include the following items when submitting your revised manuscript:

We look forward to receiving your revised manuscript.

Kind regards,

Guang-Chao Chen

Academic Editor

PLOS Genetics

Marnie Blewitt

Section Editor

PLOS Genetics

Aimée Dudley

Editor-in-Chief

PLOS Genetics

Anne Goriely

Editor-in-Chief

PLOS Genetics

**Additional Editor Comments:**

Following review by three experts, the manuscript requires further experimentation as suggested before it can be considered for publication.

**Journal Requirements:**

1) Please provide an Author Summary. This should appear in your manuscript between the Abstract (if applicable) and the Introduction, and should be 150-200 words long. The aim should be to make your findings accessible to a wide audience that includes both scientists and non-scientists. Sample summaries can be found on our website under Submission Guidelines:

https://journals.plos.org/plosgenetics/s/submission-guidelines#loc-parts-of-a-submission

2) We have noticed that you have uploaded Supporting Information files, but you have not included a list of legends. Please add a full list of legends for your Supporting Information files after the references list.

3) Some material included in your submission may be copyrighted. According to PLOSu2019s copyright policy, authors who use figures or other material (e.g., graphics, clipart, maps) from another author or copyright holder must demonstrate or obtain permission to publish this material under the Creative Commons Attribution 4.0 International (CC BY 4.0) License used by PLOS journals. Please closely review the details of PLOSu2019s copyright requirements here: PLOS Licenses and Copyright. If you need to request permissions from a copyright holder, you may use PLOS's Copyright Content Permission form.

Potential Copyright Issues:

i) Figure 5J. Please confirm whether you drew the images / clip-art within the figure panels by hand. If you did not draw the images, please provide (a) a link to the source of the images or icons and their license / terms of use; or (b) written permission from the copyright holder to publish the images or icons under our CC BY 4.0 license. Alternatively, you may replace the images with open source alternatives. See these open source resources you may use to replace images / clip-art:

4) In the online submission form, you indicated "Please email zhangchenliang@wchscu.edu.cn with requests for raw data or reagents." All PLOS journals now require all data underlying the findings described in their manuscript to be freely available to other researchers, either

1. In a public repository

2. Within the manuscript itself

3. Uploaded as supplementary information.

1) Please clarify all sources of financial support for your study. List the grants, grant numbers, and organizations that funded your study, including funding received from your institution. Please note that suppliers of material support, including research materials, should be recognized in the Acknowledgements section rather than in the Financial Disclosure

2) State the initials, alongside each funding source, of each author to receive each grant. For example: "This work was supported by the National Institutes of Health (####### to AM; ###### to CJ) and the National Science Foundation (###### to AM)."

3) State what role the funders took in the study. If the funders had no role in your study, please state: "The funders had no role in study design, data collection and analysis, decision to publish, or preparation of the manuscript."

4) If any authors received a salary from any of your funders, please state which authors and which funders.

6) Your current Financial Disclosure states, "The author(s) received no specific funding for this work."

However, your funding information on the submission form indicates receiving a fund. Please ensure that the funders and grant numbers match between the Financial Disclosure field and the Funding Information tab in your submission form. Note that the funders must be provided in the same order in both places as well.

**Reviewers' comments:**

Reviewer's Responses to Questions

Reviewer #1: In this manuscript, Zhang et al. uncover a novel mechanism by which nuclear p62/SQSTM1 facilitates the ubiquitin-independent proteasomal degradation of BMAL1, a critical regulator of circadian rhythm. Their findings demonstrate that p62 acts as a receptor for the 20S proteasome, recruiting BMAL1 for degradation within the nucleus without relying on ubiquitination or biomolecular condensates. Through molecular and cellular analyses, the authors identify key domains in both p62 and BMAL1 essential for their interaction and degradation process. Furthermore, they propose that remodeling nuclear p62 accumulation could serve as a therapeutic strategy to suppress BMAL1-associated tumor proliferation. Despite several previous reports of the pro-degradative function of p62 in the nucleus, this study provides alternative mechanistic insights into the mode of p62 action in BMAL1 quality control and their implications for circadian regulation and cancer therapy. However, the link between p62-BMAL1 interaction and BMAL1 degradation needs further evidence. Below I have some questions.

- What’s the role of p62 in circadian clock? The authors showed p62 can degrade BMAL1, a critical clock regulator. If this role of p62 is significant, wouldn’t manipulating p62 cause a circadian phenotype? Fig 1M-T seems to suggest so, but I wonder if this is the case in animal studies. As there are reports of p62 knockout animals, I’d suggest highlighting/speculating the role of p62 in circadian regulation in vivo in introduction or summarizing previous literature in the discussion.

- Following this logic, in Fig 1M-T a correlation between p62 and expression of clock genes were shown, suggesting p62 regulates circadian cycle. However, this regulation could be BMAL1-independent. I would suggest investigating whether p62 regulates the clock genes through controlling BMAL1 levels.

- In Fig2E, it seems that different p62 constructs have different expression levels. Could this be the reason for differential BMAL-1 degradation? In other words, one could argue that FLAG-p62-WT degrades BMAL1-myc better than the delta-NLS constructs because it expresses at a higher level. I’d suggest a control to address this issue.

- Fig3: Authors claim that DR deletion abolishes the phenotype of p62-driven BMAL1 degradation. At the protein level, does BMAL1-delta-DRs associate with p62? This experiment would provide direct evidence whether the DRs are required for p62-dependent BMAL1 association? Since the authors already have delta-DR mutant constructs, I suggest further investigating the interaction between BMAL1 IDR and p62 or the 20S proteasome.

- Fig4: For functional validation, does p62-deltaUBA promote BMAL-1 degradation?

- Fig4AB: it’s hard to distinguish the nuclear-cytoplasmic boundary. Image quality needs improvement. Quantification is also in need.

- Fig4AB: Authors claim “BMAL1 did not localize to p62-associated biomolecular condensates, where the proteasome is recruited to degrade substrates in a ubiquitin-dependent manner.” Is this also true in colon cancer tissue or cell line?

- Along this line, to examine whether BMAL1 degradation requires condensate formation, I recommend testing p62 constructs lacking the PB1 or UBA domains to see whether BMAL1 can still be degraded without these condensate-forming regions.

- Fig5: Authors claim that ZZ is essential for p62-BMAL1 interaction. To validate the functional relevance of this specific domain, is p62-deltaZZ able to degrade BMAL1?

- Fig6: Since ZZ associates with BMAL1, I wonder how p62-deltaZZ would affect tumor cell growth.

Minor:

- In section one of Results: authors claim “the deletion of p62 augmented the levels of several BMAL1-targeted clock transcripts, including NR1D1, PER2, CRY1, and CRY2.” What about DBP and PER1? Explanation of the negative results is required.

- Fig2KL: Can the authors provide confocal images?

Reviewer #2: The manuscript "Nuclear p62/SQSTM1 Facilitates Ubiquitin-independent Proteasomal Degradation of BMAL1" aims to elucidate the role of nuclear p62 protein in the BMAL1 degradation pathway. The authors found that p62 promotes the proteasomal degradation of BMAL1 within the nucleus, and this process is independent of ubiquitination. They further argue that nuclear p62-driven BMAL1 degradation is not dependent on biomolecular condensates. The authors also discovered that p62 acts as a receptor for the 20S proteasome, facilitating the recruitment of BMAL1 to the 20S proteasome for degradation. Overall, although this study reveals that the accumulation of p62 in the nucleus may represent a potential strategy for targeting BMAL1 to suppress tumor cell growth, some of the mechanisms have not been fully elucidated. Therefore, I do have several major/minor concerns that it needs more solid evidence to prove the author's viewpoint for publication.

1、The authors suggest that p62 regulates the expression of clock genes by modulating BMAL1 protein levels. However, in Figures 1O and 1Q, the knockout of p62 does not significantly increase the mRNA levels of DBP and PER1, and the trend appears to be independent of time. Additionally, when p62 is normally expressed, the mRNA levels of BMAL1 show a decreasing trend between 12-20 hours. This raises the question of how exactly p62 regulates the levels of BMAL1, which remains unclear.

2、The different domains of p62 may perform distinct functions in different cellular compartments. In the nucleus, p62 may interact with nuclear proteins or RNA, leading to the rearrangement or exposure of certain domains (such as the PB1 domain). The nuclear environment may induce functional changes in these domains, which in turn affect the role of p62. Does the study explore whether p62 promotes BMAL1 degradation in the nucleus is due to changes in the function of its domains caused by the nuclear environment?

3、The author's viewpoint is that the loss of p62 significantly inhibits the interaction between BMAL1 and the 20S proteasome subunit. However, we do not know what changes occur after the binding of the 20S proteasome subunit with p62, nor what the specific binding domain of BMAL1 with the 20S proteasome subunit is.

4、The article claims that BMAL1 acts as a tumor suppressor in cancers including HCC, pancreatic cancer, and tongue squamous cell carcinoma, whereas it exhibits oncogenic properties in colon and breast cancers. However, the study exclusively focuses on the BMAL1-p62 relationship in colon cancer, without experimental evidence demonstrating how p62 or p62(ΔNES) modulates BMAL1 and cell proliferation in HCC or other tumor-suppressive cancer types.

5、It has been shown that BMAL1 is a crucial protein involved in regulating circadian rhythm. If p62-mediated BMAL1 degradation is modulated, whether does p62-mediated BMAL1 degradation (either inhibition or enhancement) affect circadian rhythmicity in mice and consequently induce adverse physiological consequences?

6、Figure 6 shows that both p62 and BMAL1 protein levels are significantly increased in colon cancer patient samples, which contradicts the author's previous finding that p62 and BMAL1 overexpression promotes BMAL1 proteasomal degradation. Is the nuclear level of p62 influenced by other key proteins, and do the levels of related proteins in the nucleus also change?

7、The manuscript mentioned that p62 can promote the degradation of BMAL1 via the proteasome pathway. So, how does p62 function as a bridge to link BMAL with proteasesome? More biochemical data shoud be provided.

Reviewer #3: 1. In Figure 1, the authors provided several data shown that ectopic expression of p62 may downregulate the ectopically expressed BMAL in HEK293 and HeLa cells. Whether an enhanced protein level of p62 leads to a reduction of the endogenous amount of BMAL remains questionable. The nearly equal amount of p62 in cytoplasmic and nuclear fractions was really issued since several studies have shown that p62 is dominantly expressed in the cytosol.

2. In Figure 1, the effect of p62 knockout on BMAL-induced gene expressions of circadian rhythm genes was not well interpreted, and the related results were not appropriately concluded.

3. In Figure 2, no detectable difference between WT and NLS deleted mutants was shown in immunofluorescence analysis (IFA). The authors should specifically indicate whether the nuclear and cytosol distribution of p62 was analyzed by confocal microscopy.

4. Since the authors have already established p62 KO cells, it is rational to ask whether overexpression of p62 NLS and NES mutants alters the endogenous level of BMAL. The effect of LMB is also encouraged to assess the endogenous level of BMAL by p62.

5. For the in vivo ubiquitination in Figure 3, please utilize another tagged ubiquitin, such as HA-ubiquitin. It remains doubtful whether p62 promotes the ubiquitination of BMAL. The immunoblot shown the in vivo ubiquitination should be analyzed by more prolonged exposure.

6. In Figure 4, please provide the quantitative data shown the ratio of nuclear and cytosolic p62 and its relevance with BMAL.

7. Does a loss of p62-BMAL and p62 and 20S interactions diminish the degradation of BMAL in the nucleus? The authors should examine the effects of the interaction-interrupted mutants on BMAL stability.

8. Since the authors have already mapped the interaction domains of p62, BMAL, and 20S, the effects of interrupted interaction between BMAL and p62, and p62 and 20S on the cell proliferation of colon cancer cells in Figure 6 should be analyzed.

9. Several references were not appropriately cited.

10. Not enough information regarding experimental strategies used in each figure was not provided in the corresponding legend.

**Have all data underlying the figures and results presented in the manuscript been provided?**

Reviewer #1: Yes

Reviewer #2: Yes

Reviewer #3: Yes

PLOS authors have the option to publish the peer review history of their article (what does this mean? ). If published, this will include your full peer review and any attached files.

**Do you want your identity to be public for this peer review?** For information about this choice, including consent withdrawal, please see our Privacy Policy .

Reviewer #1: No

Reviewer #2: No

Reviewer #3: No

**Figure resubmission:**
---

## [Decision Letter · Decision Letter 1]

Dear Dr Zhang,

We are pleased to inform you that your manuscript entitled "Nuclear p62/SQSTM1 Facilitates Ubiquitin-independent Proteasomal Degradation of BMAL1" has been editorially accepted for publication in PLOS Genetics. Congratulations!

Yours sincerely,

Guang-Chao Chen

Academic Editor

PLOS Genetics

Marnie Blewitt

Section Editor

PLOS Genetics

Aimée Dudley

Editor-in-Chief

PLOS Genetics

Anne Goriely

Editor-in-Chief

PLOS Genetics

Comments from the reviewers (if applicable):

Reviewer's Responses to Questions

**Comments to the Authors:**

Reviewer #1: I have no further questions. Recommended.

Reviewer #2: I have no concerns on this revised manuscript.

Reviewer #3: Most of the issues raised by the reviewer have been addressed.

**Have all data underlying the figures and results presented in the manuscript been provided?**

Reviewer #1: Yes

Reviewer #2: Yes

Reviewer #3: Yes

PLOS authors have the option to publish the peer review history of their article (what does this mean? ). If published, this will include your full peer review and any attached files.

**Do you want your identity to be public for this peer review?** For information about this choice, including consent withdrawal, please see our Privacy Policy .

Reviewer #1: No

Reviewer #2: No

Reviewer #3: No

**Data Deposition**

http://datadryad.org/submit?journalID=pgenetics&manu=PGENETICS-D-25-00276R1

**Press Queries**

---

## [Editor Report · Acceptance letter]

PGENETICS-D-25-00276R1

Nuclear p62/SQSTM1 Facilitates Ubiquitin-independent Proteasomal Degradation of BMAL1

Dear Dr Zhang,

We are pleased to inform you that your manuscript entitled "Nuclear p62/SQSTM1 Facilitates Ubiquitin-independent Proteasomal Degradation of BMAL1" has been formally accepted for publication in PLOS Genetics! Your manuscript is now with our production department and you will be notified of the publication date in due course.

With kind regards,

Zsofia Freund

PLOS Genetics

On behalf of:
